# Radiomics in Oncology, Part 1: Technical Principles and Gastrointestinal Application in CT and MRI

**DOI:** 10.3390/cancers13112522

**Published:** 2021-05-21

**Authors:** Damiano Caruso, Michela Polici, Marta Zerunian, Francesco Pucciarelli, Gisella Guido, Tiziano Polidori, Federica Landolfi, Matteo Nicolai, Elena Lucertini, Mariarita Tarallo, Benedetta Bracci, Ilaria Nacci, Carlotta Rucci, Elsa Iannicelli, Andrea Laghi

**Affiliations:** 1Radiology Unit, Department of Medical Surgical Sciences and Translational Medicine, Sapienza University of Rome-Sant’Andrea University Hospital, Via di Grottarossa, 1035-1039, 00189 Rome, Italy; damiano.caruso@uniroma1.it (D.C.); michela.polici@uniroma1.it (M.P.); marta.zerunian@uniroma1.it (M.Z.); francesco.pucciarelli@uniroma1.it (F.P.); gisella.guido@uniroma1.it (G.G.); tiziano.polidori@uniroma1.it (T.P.); federica.l2005@libero.it (F.L.); matteo.nicolai@uniroma1.it (M.N.); elena.lucertini@uniroma1.it (E.L.); benedetta.bracci@uniroma1.it (B.B.); ilaria.nacci@uniroma1.it (I.N.); carlotta.rucci@uniroma1.it (C.R.); elsa.iannicelli@uniroma1.it (E.I.); 2Department of Surgery “Pietro Valdoni”, Sapienza University of Rome-Umberto I University Hospital, Viale del Policlinico, 155, 00161 Rome, Italy; mariarita.tarallo@uniroma1.it

**Keywords:** Radiomics, oncologic imaging, Radiomics technical principles

## Abstract

**Simple Summary:**

Part I is an overview aimed to investigate some technical principles and the main fields of radiomic application in gastrointestinal oncologic imaging (CT and MRI) with a focus on diagnosis, prediction prognosis, and assessment of response to therapy in gastrointestinal cancers, describing mostly the results for each pre-eminent tumor. In particular, this paper provides a general description of the main radiomic drawbacks and future challenges, which limit radiomic application in clinical setting as routine. Further investigations need to standardize and validate the Radiomics as a helpful tool in management of oncologic patients. In that context, Radiomics has been playing a relevant role and could be considered as a future imaging landscape.

**Abstract:**

Radiomics has been playing a pivotal role in oncological translational imaging, particularly in cancer diagnosis, prediction prognosis, and therapy response assessment. Recently, promising results were achieved in management of cancer patients by extracting mineable high-dimensional data from medical images, supporting clinicians in decision-making process in the new era of target therapy and personalized medicine. Radiomics could provide quantitative data, extracted from medical images, that could reflect microenvironmental tumor heterogeneity, which might be a useful information for treatment tailoring. Thus, it could be helpful to overcome the main limitations of traditional tumor biopsy, often affected by bias in tumor sampling, lack of repeatability and possible procedure complications. This quantitative approach has been widely investigated as a non-invasive and an objective imaging biomarker in cancer patients; however, it is not applied as a clinical routine due to several limitations related to lack of standardization and validation of images acquisition protocols, features segmentation, extraction, processing, and data analysis. This field is in continuous evolution in each type of cancer, and results support the idea that in the future Radiomics might be a reliable application in oncologic imaging. The first part of this review aimed to describe some radiomic technical principles and clinical applications to gastrointestinal oncologic imaging (CT and MRI) with a focus on diagnosis, prediction prognosis, and assessment of response to therapy.

## 1. Introduction

Radiomics is becoming central in emerging precision medicine, in particular for oncologic patients. Nowadays, conventional imaging is shifting from qualitative to quantitative approach, especially in tumor diagnosis, prognosis prediction, and assessment of response to therapy [1]. Radiomics has the expectancy to be an additional tool to provide a quantitative evaluation of tumor in a manner complementary to the observational approach and can be considered a helpful tool for physicians to manage oncologic patients by building a structured and objective workflow.

Radiomic parameters are ultrastructural quantitative data extracted from specific regions of interest (ROIs) [2] selected on encrypted medical images, that reflect neoplasm phenotypes and heterogeneity, usually correlated with tumor aggressiveness [2,3,4]. Then, radiologists, in a single image evaluation, can obtain an estimation of volumetric tumor heterogeneity in terms of radiomic parameters (i.e., tumor shape and textural parameters). In such scenario, oncologists may have an additional non-invasive biomarker to complement the biopsy that often can result too reductive or not diagnostic [5]. In addition, cancer biopsy could have several limitations related to risks for the patients (i.e., internal bleeding or pneumothorax), poor reproducibility, bias of random sample, limited accuracy in cancer grading and tumor invasion, and difficulty to sample some lesions [6,7]. Radiomics could be a supporting tool for cancer biopsy, a cornerstone of oncological management by guiding to the site with the most representative lesion heterogeneity, helpful for a better choice for biopsy or direct surgical resection rather than random sampling, and providing some additional information regarding tumor biology [8].

This recent landscape of imaging has been attracting the attention of many researchers, driven by promising results achieved through integration of Radiomics with clinical biomarkers. Song et al. reported that 553 original articles concerning Radiomics were published in the recent years [9]. The aim of this paper is to provide an overview of technical principles and the major radiomic studies that focused on diagnosis, prognosis assessment, and response to therapy in cancer patients, through an accurate description of prominent results in the most diffuse types of cancer with clinical significance. In Part 1, we will focus our attention on gastrointestinal applications.

## 2. Technical Principles

Radiomics workflow is divided into multi-steps: images acquisition, image segmentation, features extraction, features selection, model construction, and validation. Each step is dependent on the previous one and the aim is to obtain a performant and reliable prognostic model, usually driven by Artificial Intelligence, a research field in which the human intelligence is mimicked by mathematical and statistical approaches that are able to create performant artificial neural networks (Figure 1) [10]. Machine learning, neural network, and deep learning are some Artificial Intelligence subfields used for applying mathematical and statistical approaches to Big Data (e.g., radiomic features, clinical data, disease free survival, and overall survival) with the ability to find and interpret occult models, built on mineable data and their interpretation needed for clinical support [11]. To standardize radiomic approach in routine clinical setting, a structured and robust workflow is necessary, through which reliable and consistent data could be obtained.

The first step is based on images acquisition, which is one of the most challenging aspects because of the lack of protocol and parameters’ standardization and the fact that it affects the reproducibility of analysis, particularly important in multicenter center studies [12]. Several options have been recently proposed to overcome the bias of acquisition protocols by performing a test–retest analysis, with the goal of eliminating radiomic features affected by higher variability and maintaining only robust features [13]. One of them is an automatic acquisition protocol that has been proposed among several CT scanners, and these methods showed some promising results in terms of reaching robustness in radiomic parameters by using resampling approach to uniform each voxel size as post-acquisition correction. However, this approach has not been able to modulate acquisition specifications for each different CT scanner at the same time, and it could be difficult to use in retrospective studies routinely [14]. Regarding test–retest analysis, some relevant results have been obtained by analyzing the reproducibility of radiomic parameters among different CT scanners, both by changing CT specifications (intra-analysis) and by comparing different CT scanners (inter-analysis), and it was shown that high number of radiomic parameters could be altered by changing of some CT acquisition parameters [15]. However, these results were performed in phantom studies, in which the inter-patient variability has not been assessed; thus, future investigations are needed to identify the best approach to standardize the image acquisition. Nevertheless, the robust features often resulted to have low diagnostic performance [16]. In addition, applying a post-reconstruction batch harmonization has been also proposed to reduce the variability among centers by using global scaling, in which signal intensities are harmonized by eliminating the mean and the unit variance is downsized; z-standardization, where each feature is normalized considering the mean and standard deviation by providing some comparable results; and histogram-matching, by transforming intensity histogram in order to combine them and find the reference histogram [17]. However, each method has different advantages and drawbacks, and they should be weighted in different clinical scenarios.

The second phase is image segmentation, in which ROIs are outlined around cancer tissues, by covering the entire lesion area and avoiding some unnecessary structures (i.e., vessels, biliary duct, healthy parenchyma) that could alter the heterogeneity analysis. However, for very tiny non-lesion structures it is not always achievable, and this might represent a source of bias. The image segmentation is the leading cause of features variability, with a specific focus on inter-reader lack of reproducibility due to high variability of extracted features without a specific feature selection. High variability could affect the consistence of radiomic signature; thus, the reproducible features should be selected, excluding features classified as consistently unstable. Interclass correlation coefficient (ICC) might be used to evaluate the robustness of radiomic features among several datasets; in fact, there is evidence that a radiomic feature with high ICC on one dataset will also be robust and stable in other dataset [18,19]. Radiomic parameters are extracted from image volumes by performing manually, semi-automatic, or completely automatic segmentation. The first method is a time-consuming process, usually affected by high intra- and inter-observer variability and also according to radiologist’s experience, and is difficult to apply in a clinical setting as a routine practice [20]. Semi-automatic and automatic segmentation process are demonstrated to be promising in homogeneous lesions, where interaction of external reader was minimum, with high accuracy, low inter-reader variability, and high reproducibility [21]. Since radiomic analysis could be highly affected by these different methods of segmentation, validation, standardization, and robustness are needed [22,23,24]. Consequently, the assessment of radiomic features variability related to segmentation processes has been emerging with the proposal to eliminate the features with high variability and low prognostic strength.

Feature extraction represents the third step of radiomic process. Quantitative features have to be extracted from ROIs previously outlined on tissues of interest (Figure 2). Features obtained are divided into shape features, describing the shape and geometry of ROIs (i.e., volume, maximum surface): first-order statistics features, resulting from gray-level histogram and describing voxel values without considering the relationship with other voxels; second-order statistics features (i.e., gray-level co-occurrence matrices or gray-level run length matrices), derived by analyzing each pixel and its relationship with those adjacent in specific matrices; and high-order statistics features, resulting from mathematical algorithm after the application of specific filters (Table 1).To avoid some unnecessary bias, affecting data homogeneity, it is critical to standardize each step of analysis [25].

Feature selection is one of the main key steps of radiomic process. It is related to the selection of best performing parameters among the large quantity of parameters extracted, often interconnected and characterized by overfitting. In that context, a specific and punctual selection of obtained features seems to be necessary to avoid some bias in model construction; it is essential to define the endpoint of analysis with precise clinical applications in order to select the best performing features with some dedicated approaches. Two most unsupervised approaches that are used to perform features selection are cluster analysis and principal component analysis. The first is based on clustering similar radiomic parameters according to high cluster redundancy and low inter-cluster correlation, usually illustrating as cluster heat map; and only one parameter from each cluster is selected for further analysis. The principal component analysis is built on creating a small set of non-correlated features extracted from a large amount of correlated features with the aim to explain the total variable variation using the smallest number of features, usually illustrating as score-plots [1]. The goal is to select the best features assessed as non-redundant, reproducible, and performing.

The next step is the model building in which the best selected features, clinical data, and histological data are combined to assess the pre-fixed outcome (e.g., survival, disease free progression, and therapeutic assessment). In this critical step, several approaches have been proposed to perform multivariate analysis, tailored per endpoint, which could provide a clinical tool able to support clinical decision-making. Different statistical methods and data mining and/or machine-learning methods were investigated. The main classifiers used in practice are random forest, linear regression, logistic regression, Cox proportional hazards regression, least absolute shrinkage and selection operator (LASSO), support vector machines (SVM), neural network, deep learning, and decision tree [26]. To date, it was shown that there is no unique best classifier, but each of them should be evaluated in order to obtain the consistent, reliable, and generalizable models [27].

In conclusion, the constructed model must be trained and optimized on training and testing set, then validated on external cohort in order to obtain a reliable model on different patient groups. Although the validation on external datasets is necessary, sometimes it could be impossible. Hence, several strategies have been tested to overcome this limitation by obtaining an internal validation (i.e., random subsampling and nested approach) but the main issue of these approaches is the risk to alter the features selection algorithm, and the results obtained could be unreasonably optimistic [28]. To sum up, the validation model achieved through external validation is the goal standard and the comparability of radiomic features extracted from medical images, acquired with different protocols, and segmentation processes performed with different tasks are the main challenges of radiomic approaches.

## 3. Esophageal Cancer

Several studies investigated Radiomics in staging, response to therapy prediction, and post-operative recurrence of esophageal squamous cell carcinoma (ESCC). Regarding ESCC staging, Wu et al. [29] proposed a radiomic approach, based on computed tomography (CT) late arterial phase, by performing manual segmentation. Ten radiomic features were selected to build the signature, and the radiomic model resulted to be superior to tumor volume, a morphological characteristic reflecting tumor size, in distinguishing early (I–II) from advanced stage (III–IV), both in primary (AUC 0.795 vs. 0.694, respectively) and validation cohort (AUC 0.762 vs. 0.624, respectively). Radiomic approach was also confirmed by Liu and colleagues [30], who proposed CT Texture Analysis (CTTA) on baseline CT as a possible biomarker to evaluate ESCC aggressiveness (T, N, overall stages, and grading differentiation) in the preoperative clinical setting. Kurtosis and entropy were found as independent predictors for T stages (T1–2 vs. T3–4) and overall stages (I–II vs. III–IV); in addition, entropy showed the best performance in terms of T stages (T1–2 vs. T3–4), lymph node metastasis (N− vs. N+), and overall stages (I/II vs. III/IV) with AUC 0.637, 0.815, and 0.778, respectively.

Radiomics also resulted to be promising in the pretherapeutic assessment of chemotherapy response [31,32,33]. Jin et al. [31] tested an integrated model of CT radiomic features using machine learning approach in 94 patients affected by esophageal cancer who underwent neoadjuvant chemotherapy (CRT) (58 responders and 36 non-responders). A total of 42 radiomic features were extracted, and the integrated radiomic and dosimetric model obtained a good accuracy (AUC 0.708) in predicting the treatment response of CRT in patients with ESCC. Differently, Nakajo and colleagues [33] investigated the heterogeneity of F-18-fluorodeoxyglucose (18F-FDG) distribution in predicting tumor response in 52 patients affected by esophageal cancer. Despite textural parameters and volumetric features correlating with tumor response, texture analysis did not show a relevant role in estimation of esophageal cancer prognosis. Among treatment assessment, Hu and colleagues [32] tested Radiomics to evaluate tumor complete response in 231 patients with locally advanced ESCC treated with neoadjuvant CRT, by combining intratumoral and peritumoral features, obtaining an AUC of 0.852.

Furthermore, several studies focused on predicting risk of pathological lymph nodes (LN) in esophageal cancer patients such as that of Wu et al. [34] and Shen et al. [35] in which the authors proposed Radiomics as a tool in pretherapeutic LN assessment. A Radiomic signature was built from 13 radiomic features extracted from 197 ESCC patients, and it was significantly associated with pathological LN (*p* < 0.001). Moreover, the AUC was 0.806 in the training model and 0.771 in the validation model, suggesting that Radiomics may be helpful for suspected LN.

Recently, radiomic application in recurrency has been investigated by Qiu et al. [36] validating a radiomic combined nomogram (quantitative and clinical parameters) based on preoperative CT scans of 206 ESCC patients treated with neoadjuvant CRT and surgery; the model showed high correlation with recurrence-free survival both in training (AUC 0.746) and validation cohort (AUC 0.724).

## 4. Gastric Cancer

In the last few years, several authors have also investigated Radiomics in gastric cancer (GC) management, and the major results were obtained in diagnosis and staging, prognosis stratification, and treatment response prediction. In a preoperative clinical setting, Radiomics was applied to evaluate histopathological tumor characteristics, and to predict risk of nodal and peritoneal metastases [37,38,39]. Liu and colleagues studied CTTA to evaluate histopathological findings in 392 pre-operative CT scans by using endoscopic biopsy as the reference standard [37]. CTTA was performed on arterial and venous phase, and each ROI was manually drawn on selected slice where cancer area was depicted. Authors made a comparison between CTTA features and histological data. They obtained that mean attenuation, maximum attenuation, all percentiles, and mode extracted from CT venous phase correlated with tumor differentiation degree (from r = −0.231 to r = −0.324, *p* < 0.017), and hence CTTA could be helpful to identify well differentiated cancers (*p* < 0.004). This radiomic approach was reinforced by Wang et al. [38], who obtained some relevant results by testing a CT-based radiomic nomogram (radiomic features + clinical parameters) to predict pathological LN before any therapy. By using a random forest algorithm, radiomic model showed a good power in discriminating pathological from non-pathological LN with an AUC of 0.844 and 0.837 for training and test population, respectively, while the radiomic nomogram showed better results for both training and test cohort with an AUC of 0.886 and 0.881, respectively.

Surprising results were obtained by Dong and colleagues [39] aimed at using a radiomic approach in identifying occult peritoneal metastases (PM) in a multicentric study listing 554 patients affected by gastric cancer in pretherapeutic setting. All patients have a negative CT for peritoneal involvement while laparoscopy confirmed the presence of PM in 122/554 patients. In total, 266 quantitative radiomic features were extracted from primary tumor and from peritoneum region, and the predicting power of the derived radiomic nomogram was surprising with an AUC of 0.958, 0.928, and 0.920 for training and two external validation datasets, respectively.

Recently, Giganti et al. [40] proposed CTTA to estimate pretherapeutic overall survival (OS) in 56 patients with resectable GC by performing a manually volumetric segmentation of entire tumor volume. Authors revealed that some textural parameters of heterogeneity, extracted from preoperative CT scans, were strictly linked to poor prognosis (*p* < 0.046). Furthermore, Li et al. [41] confirmed these promising results of Radiomics in GC prediction of prognosis by analyzing 181 GC patients who underwent radical resection and then analyzing radiomic signature alone and combining clinical–pathological data. Their results revealed that radiomic integrated nomogram had a higher prognostic value in stratifying patients into high- and low-risk of death (Harrel concordance index, 0.82 vs. 0.74, *p* < 0.001).

Only few studies investigated Radiomics in pretherapeutic prediction of chemotherapy response. Jiang et al. [42] developed a radiomic score to evaluate stage II and III in patients that could benefit from adjuvant chemotherapy. They divided patients into low, medium or high radiomic score categories based on 1147 baseline CT, and a comparison was made between disease-free survival (DFS) and OS of patients who received chemotherapy and patients who did not. Results showed that the advantage of adjuvant chemotherapy for patients with high radiomic score was higher (HR < 0.412, *p* < 0.001) than patients with low or medium score. Similar and concordant results were obtained by Sun et al. [43], which validated radiomic and clinical scores based on 106 CT (74 training and 32 validation cohort) of patients at III and IV clinical stage in order to predict neoadjuvant chemotherapy (NAC) response. They showed higher performance of Radiomics (AUC 0.77 training cohort; AUC 0.82 validation cohort) in comparison with only clinical score (AUC 0.70 training cohort; AUC 0.62 validation cohort); however, they did not obtain statistically significant differences between two scores (*p* = 0.15).

## 5. Liver and Biliary Tract

The liver cancer workup has been extensively debating, and hepatocellular carcinoma (HCC) diagnosis represents the challenge of imaging in cirrhotic patients having several indeterminate nodules. In this context, Radiomics was recently proposed as an additional tool to identify HCC nodule and assess HCC grading [44,45]. Mokrane et al. [44] extracted CT quantitative parameters from 178 cirrhotic patients in order to identify HCC among indeterminate nodules, in the light of optimizing patient management and improving outcome by reducing the rate of liver biopsy or wait-and-see strategy in fragile patients. No significant results were reported by comparing HCC and non-HCC nodules using LI-RADS scores, while radiomic signature, based on delta quantitative features (between venous and arterial phases), reached an AUC of 0.70 in training cohort (142 patients) and 0.66 in validation cohort (36 patients).

Wu and colleagues [45] aimed to investigate magnetic resonance imaging (MRI) radiomic signature to distinguish low- from high-grade HCC. The authors showed relevant results to differentiate two patient groups (*p* < 0.05), combining radiomic and clinical features, with an AUC of 0.800. Furthermore, both radiomic signature and alpha fetoprotein level resulted to be factors independent of HCC grading (*p* < 0.05).

Regarding cholangiocarcinoma diagnosis, Yang et al. [46] retrospectively performed a radiomic analysis on preoperative MRI images in 100 patients with pathological diagnosis of extrahepatic cholangiocarcinoma aiming at detecting both differentiation degree (high- and medium-low differentiation) and nodal status in pretherapeutic clinical setting. A radiomic model was built incorporating radiomic features and ADC value obtaining significant results in differentiation degree (AUC 0.78) and in nodal status prediction (AUC 0.80).

Regarding OS, Kim et al. [47] proposed three scores (radiomic, clinical alone, and combined) to assess OS in patients affected by HCC and treated with trans-arterial chemoembolization showing the combined score as the better survival predictor with a HR of 19.88 (*p* < 0.0001). Microvascular invasion (MVI) was shown to be one of the main predictors of outcome and tumor grading in HCC patients, and more aggressive surgery could be planned in case of MVI prediction. To date, preoperative detection still represents an imaging challenge compared with histology as a reference standard. Recently, high interobserver variability has also been shown in MVI evaluation on MRI among expert radiologists, showing poor performance [48]. In such scenario, Radiomics could be a key tool as shown by Xu et al. [49], who created and tested a radiographic–radiomic, CT-based model, including clinical, radiomic, and radiographic parameters, and obtained a high precision in predicting pretherapeutic MVI (AUC 0.88). Ni and colleagues [50] explored a performant diagnostic model to predict MVI on CT scans, obtaining substantial results in agreement with a previous study. In total, 1044 textural features were extracted from CT in 88 MVI-positive and 118 MVI-negative patients. Twenty-one radiomic methods were tested, showing promising results for the LASSO method combined with gradient boosting decision tree which reached the best performance in distinguishing MVI-positive and MVI-negative (accuracy 84%, sensitivity 82%, specificity 85%, and AUC 0.88). Among MRI, Feng et al. [51] developed and validated a radiomic model in 110 HCC patients before surgery; intratumoral and peritumoral radiomic features were extracted, and the resulting model demonstrated good performance in MVI prediction with AUC of 0.85 both in training and validation cohorts.

Radiomics was also extensively studied to predict the risk of recurrence, both in HCC and in cholangiocarcinoma [52,53]. Radiomic and machine learning approach were extensively investigated by Ji et al. [52] to predict HCC recurrence in 470 patients, who underwent pretherapeutic CT and surgical resection of solitary nodule. The authors developed and validated two models, combining radiomic and clinical factors, that were able to predict risk of recurrence pre- and post-surgery. A three-feature signature was developed obtaining favorable results in predicting HCC recurrence (C-index of 0.699). Similarly, Liang et al. [53] conducted a single-center retrospective study developing a radiomic nomogram, based on pretherapeutic arterial-phase contrast-enhanced MRI, in order to predict early recurrence of intrahepatic cholangiocarcinoma after primary surgery. Radiomic combined model reached a better performance to predict risk of recurrence after surgery with AUC of 0.9 and 0.77 in training and validation cohorts, respectively.

In conclusion, liver metastases was an interesting research field in colorectal cancer, where texture features have shown a remarkable value as imaging biomarker, capable of differentiating patients with high- and low-risk of developing synchronous or metachronous liver metastases [54]. Taghavi et al. [54] concluded that a hybrid model, combining clinical and texture parameters, achieved the best prediction performance, yielding an AUC of 86% in predicting the occurrence of liver metastases and demonstrating that a non-invasive, artificial-intelligence-based model could support individualized therapy and improve oncological outcome.

## 6. Pancreas

Application of Radiomics in pancreatic cancer was mostly studied to diagnose pancreatic ductal adenocarcinoma (PDAC), to assess response to therapy, and to predict overall survival (Figure 3). In particular, Radiomics was investigated to overcome CT scans’ limitations in differential diagnosis between PDAC and pancreatitis, which could often mimic cancer [55,56,57,58]. Ren et al. [55] applied Radiomics on CT to differentiate PDAC from mass-forming pancreatitis. The authors retrospectively enrolled 109 patients (79 with PDCA and 30 with mass-forming pancreatitis) who underwent unenhanced CT prior to surgery. Volumetric Radiomics was extracted from CT images and then diagnostic performance was assessed considering histology as a reference standard. Among 396 radiomic features extracted, the four most significant predictive parameters were considered to build a predictive model with a high accuracy, sensitivity, and specificity (93.3%, 92.2%, and 94.2%, respectively). Since then, other differential diagnosis evaluated with Radiomics included autoimmune pancreatitis and healthy pancreatic parenchyma. In fact, Park and colleagues [57], reinforcing the previous study, performed a CT-machine learning method to differentiate autoimmune pancreatitis from PDAC. Autoimmune pancreatitis is an entity pathologically distant from PDAC but visual imaging features have an important overlap, making difficult both the discrimination between the two and the consideration of different treatment options. Two types of CT Radiomics were extracted based on different CT phases (arterial and portal-venous phases) and slice thickness (thin and thick slices). The study population was divided into test and validation cohorts and a random forest classifier was applied to select the most significant features. CT Radiomics extracted from thin slice portal-venous phase showed the correct classification of 100% (33/33) of the PDAC included in the validation cohort, with sensitivity, specificity and accuracy of 89.7%, 100%, and 95.2%, respectively. On the other hand, Chu et al. [56] tested the Radiomics’ ability to identify PDAC from normal pancreatic tissue, performing 3D manual segmentation of the whole pancreatic parenchyma, assessed on portal-venous phase. One-hundred and ninety PDAC and 190 healthy patients were analyzed, and Radiomics yielded an accuracy of 99.2% (AUC 0.99, sensibility 100%, and specificity 98.5%) in the diagnosis of PDAC cases.

Radiomics was also tested in the evaluation of pretherapeutic response to CRT by Nasief et al. [59], who investigated the predictive value of delta-Radiomics on 28 unenhanced CT scans obtained during therapy. Good performance resulted in distinguishing good from bad responders (AUC 0.94). Similar results were obtained by Parr et al. [60], who developed a CT-based radiomic model to assess the clinical outcome in 74 patients who underwent stereotactic body radiotherapy. The radiomic-model achieved a better performance (AUC 0.78) than clinical model (AUC 0.66) in differentiating high- from low-risk patients.

Furthermore, several studies investigated Radiomics as non-invasive biomarker in prognosis and OS prediction in PDAC. Eilaghi et al. [61] applied CT texture analysis (CTTA) on patients with resectable PDAC to assess OS and achieved a statistically significant correlation (*p* < 0.05 and AUC 0.72) between OS and two textural parameters (dissimilarity and inverse difference normalized). Recently, Kim et al. [62] proposed to test CTTA to measure OS in PDAC patients, who underwent surgery after NAC. They calculated delta textural features between pre-surgery and baseline CT images and found a statistically significant correlation between CTTA features and OS (higher subtracted entropy, *p* = 0.005 and HR = 0.159, and lower subtracted GLCM entropy, *p* = 0.036 and HR = 10.325). In addition, Tang et al. [63] developed a radiomic MRI-nomogram based on 303 patients with resectable PDAC to stratify early recurrence risk. Combining radiomic and clinical parameters, they showed an AUC between 0.88 and 0.85 depending on training and internal and external validation cohorts.

## 7. Small Bowel

Small bowel neoplasms are rare entities, and early-stage small bowel tumors are difficult to recognize on conventional imaging. Even if esophagogastroduodenoscopy with endoscopic ultrasound is recommended during initial workup, CT or MRI may be useful to evaluate the extent of local tumor invasion and to evaluate distant metastases [64].

Currently, radiomic-based models that are able to diagnose and characterize small bowel tumors are a few in number, mostly dedicated to a specific histotype of small bowel tumors. In particular, Lu and colleagues [65] performed a first-order histogram-based analysis on 74 patients to differentiate duodenal adenocarcinoma (DAC), PDAC, and gastrointestinal stromal tumor (GIST) of periampullary region. Whole-lesion CT histogram analysis demonstrated significant differences among DAC, PDAC, and GIST (all *p* < 0.05), whereas performance analysis reported significant AUC for the 90th percentile on portal-venous phase for differentiating DAC from PDAC (AUC 0.854, sensitivity 92.3%, and specificity 80%), while the 90th percentile on arterial phase was the most accurate in the differentiation between DAC and GIST (AUC 0.809, sensitivity 100%, and specificity 64%) and between PDAC and GIST (AUC 0.936 sensitivity 90%, and specificity 86%).

Interesting studies on GIST have recently been performed for risk stratification on baseline CT [66,67,68], about DFS and disease progression after surgery [69] or after therapy with tyrosine kinase inhibitors [70]. For example, Ren [66] and colleagues validated a nomogram based on Radiomics and CT signs together, to preoperatively predict malignancy in 440 GIST patients. Interesting results in differentiating low-malignant from high-malignant GIST were reported by them, and the radiomic nomogram was identified as an excellent discriminator with AUC of 0.935 and 0.933 for training and validation datasets, respectively. These encouraging results might be useful in the future to build robust decision models.

## 8. Neuroendocrine Tumors

Radiomics was recently applied to manage neuroendocrine tumors (NETs), extremely rare tumors. Few studies proposed a Radiomics approach in pancreatic neuroendocrine tumors (PNETs) to assess histological degree, differential diagnosis, and prognosis. In particular, Guo et al. [71] tested CTTA ability to differentiate PNETs G1/G2 from G3, by performing a manual 2D primary lesions’ segmentation on baseline contrast-enhanced CT. Some textural parameters differ between two patient groups with good sensitivity (73–91%) and specificity (85–100%). Similar and concordant results were proposed by Gu and colleagues [72], who performed a CT-radiomic nomogram, combining quantitative and clinical factors, to predict PNETs grading (G1 and G2/G3) on 138 patients, who underwent contrast-enhanced CT at diagnosis. They demonstrated that the combined model was extremely promising to differentiate PNET G1 from G2/G3 with AUC 0.97 in training and 0.90 in validation cohort. In addition, Bian and colleagues [73] performed a radiomic model to evaluate preoperative tumor grading in patients affected by non-functional PNETs on 3T MRI and confirmed promising radiomic results. They visually evaluated non-functional PNETs for radiological findings used for the clinical score and then selected a volume of interest (VOI) of the lesion to extract radiomic features. Both clinical model and radiomic model were obtained; results were obtained on 97 patients and then validated on 42 patients. An AUC of 0.769 and 0.72 (training and validation group, respectively) was reached by the clinical model including 14 imaging features, while 14 radiomic features showed good discrimination in the training cohort (AUC 0.851) and the validation cohort (AUC 0.736).

He and colleagues [74] built and validated a CT-based radiomic model to differentiate atypical non-functional PNETs from PDAC, by testing radiomic features alone and in combination with clinic-radiological features. Radiomics was extracted from preoperative enhanced CT of 147 patients. Three models (only clinical–radiological, only Radiomics, and combined) were trained and validated to differentiate atypical non-functional PNET from PDAC. The clinical–radiological model showed an AUC of 0.775, while the radiomic model reached an AUC of 0.873. The combined model resulted in a slightly higher performance than radiomic model, with an AUC of 0.884.

Finally, regarding differential diagnosis among NETs, Martini et al. [75] evaluated ability of CTTA, extracted from liver metastases, to differentiate pancreatic and non-pancreatic NETs with the same grading (G1 or G2). Volumetric analysis on liver metastasis was performed on 48 patients (23 with PNET and 25 with non-pancreatic NET). Their results showed how first-order Radiomics provided some relevant differences in differentiating PNETs from non-pancreatic NETs in both arterial phase (skewness feature, G2) and portal-venous phase (mean feature, for G1 and G2), all *p* < 0.05.

In addition, interesting correlation was also found regarding Radiomics with OS (mean feature extracted from both arterial and portal-venous phase, r > 0.42) and with time to progression (entropy feature extracted from arterial phase, r < −0.42).

No differences emerged to discriminate tumor lesions for grading (G1 vs. G2); as well as no significant correlation were observed between Radiomics and Ki-67 in both PNETs and non-pancreatic NETs groups.

## 9. Colorectal Cancer

Radiomics and Radiogenomics were also widely investigated as imaging biomarker in colorectal cancer (CRC) in the assessment of mutational status, nodal metastases, stratification patient risk, and in evaluation of response to therapy [76,77,78,79,80,81]. Regarding CRC Radiogenomics, Yang et al. [82] retrospectively investigated whether CT-based radiomic signature could predict KRAS, NRAS, and BRAF mutations in CRC by analyzing a primary cohort (61 patients) and a validation cohort (56 patients). They made a correlation between radiomic features, clinical data, and genetic mutations and reported a significant (*p* < 0.001) association between mutational status and the proposed radiomic signature (AUC 0.869, sensitivity 0.857, and specificity 0.833). No significant results were obtained in the comparison between clinical/histological data and genetic profiling (*p* > 0.05). In addition, multifunctional imaging has been tested for CRC genotyping; Taguchi et al. [83] compared the CTTA with Maximum Standard Uptake Values (SUV_max_) performance from ^18^F-FDG PET/CT, and concluded that Radiomics was superior in distinguishing between wild type and mutant KRAS colon cancers (AUC 0.82 vs. 0.58) in agreement with the previous study. Golia Pernicka et al. [84] enhanced Radiogenomics results in prediction prognosis and developed some models based on clinical and radiomic parameters, and compared their performances in the assessment of microsatellite instability (MSI); the combined clinical–radiomic model achieved the best results in predicting MSI (AUC 0.79, specificity 92.5%, sensitivity 31.6%, positive predictive value (PPV) 66.7%, and negative predictive value (NPV) 74%).

Li et al. [85] tested Radiomics as a pretherapeutic tool in the investigation of pathological LN before starting any therapy. Six predictive models were compared, and the most relevant results were obtained by combining clinical variables and LN-radiomic features, extracted from primary tumor and peripheral nodes in 458 patients of primary cohort and 308 of validation cohort (AUC 0.75, accuracy 73%, sensitivity 60%, and specificity 84% in testing set). Regarding rectal cancer, in order to assess pretherapeutic nodal status by employing textural features extracted from primary tumor on baseline MRI scans, Radiomics has been reported to range from 50% to 93% for sensitivity and from 43% to 80% for specificity in the latest studies [86,87] with a good interobserver agreement reproducibility in clinical practice, confirming the relevant results previously described.

Recently, a radiomic classifier has been tested to assess the response to treatment demonstrating a better performance compared with traditional evaluation (T2w and DWI), showing an AUC of 93% in distinguishing complete from partial responder patients, affected by rectal cancer, showing sensitivity, specificity, PPV, and NPV of 100%, 91%, 72%, and 100% respectively, (vs. 84%, 56%, 94%, and 30% achieved with T2w-DWI qualitative assessment) [88].

## 10. Limitations GI Radiomics

To date, Radiomics has several relevant drawbacks that limit the routine use in the clinical management of gastrointestinal cancer patients. The major radiomic limitations are the lack of standardization, both in image protocols of acquisition and in methods of segmentations; the lack of validation in prospective and large sample of patients, enrolled from multicentric studies; and an undefined correlation with histological diagnosis. These aspects could affect the reliability and stability of radiomic features, especially when extracted from small sample of patients, in which the risk of oversampling has to be considered. In particular, the application of Radiomics in differential diagnosis between benign and malignant lesions needs to be validated and compared with histological results, considered as a reference standard. In addition, nodal metastases and genetic panel prediction has to be further validated in larger patient populations. Nevertheless, the application of Radiomics has been shown to be a promising tool in each step of cancer patient’s workup, able to support clinical approach to tailor treatment per patients. In the future, Radiomics should be investigated and validated in a more structured manner to overcome the major limitations.

## 11. Conclusions

Up to now, Radiomics is an emerging and promising imaging tool for oncologist in the development of personalized cancer medicine. Quantitative approach in medical imaging has been overcoming the traditional subjective evaluation, resulting to be more promising in the assessment of tumor aggressiveness and patient’s survival. Radiomics has been extensively studied in the management of cancer patients as a complementary tool for the clinicians, and the major results were obtained by combining clinical and quantitative data. In Part I, the main radiomic technical challenges were the lack of standardization in images acquisition, segmentation, extraction, and processing features. Actually, Radiomics resulted to be absolutely promising in diagnosis, prediction prognosis, overall survival, and risk of recurrence, representing the future application of radiology. However, it will be necessary to investigate the feasible use of Radiomics in clinical practice by providing structured models in gastrointestinal oncologic imaging to overcome the lack of repeatability and reproducibility.

## Figures and Tables

**Figure 1 cancers-13-02522-f001:**
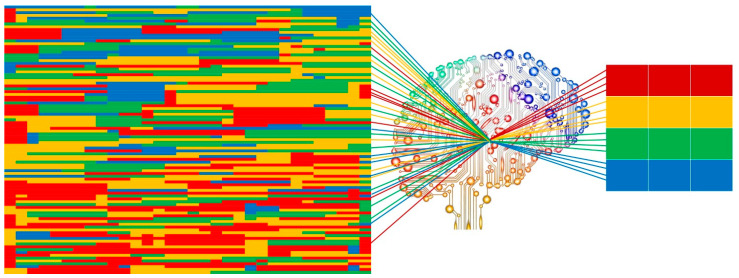
Graphic insight of Artificial Intelligence image analysis process. From imaging disorganized data to bright data.

**Figure 2 cancers-13-02522-f002:**
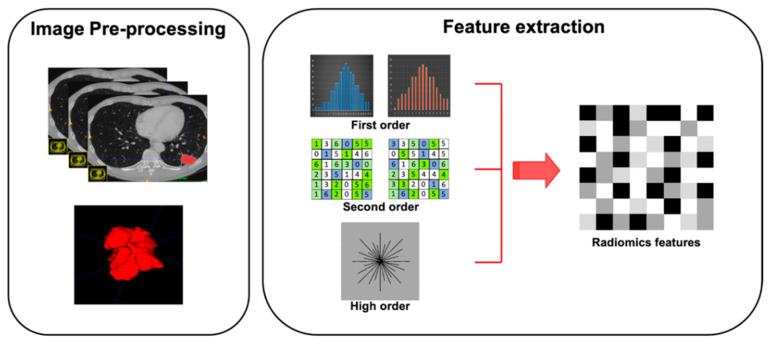
Graphical radiomic flowchart of features extraction. First order: description of voxel values without consider the relationship with others; second order: analysis of each pixel and its relationship with the adjacent pixels; higher order: result of mathematical algorithm performed with specific filters.

**Figure 3 cancers-13-02522-f003:**
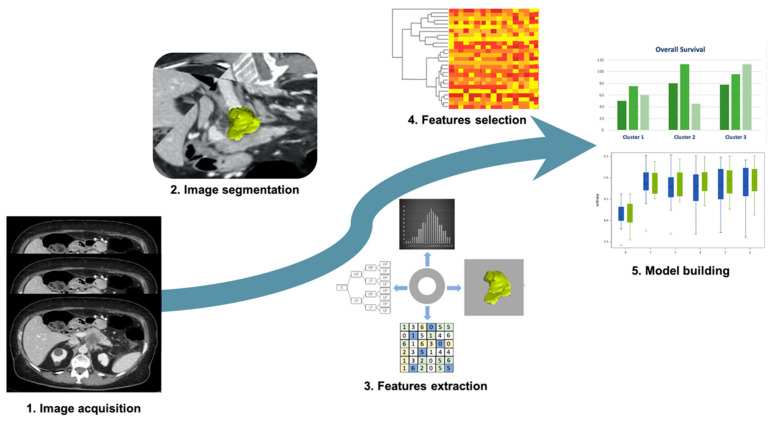
Radiomic approach on pancreatic ductal adenocarcinoma. 1. Image acquisition: CT scans acquired on venous phase; 2. Image segmentation: semiautomatic segmentation performed on CT scans; 3. Feature extraction: tumor shape, first order features, texture features, and wavelet filter; 4. Feature selection: cluster analysis was used for the features selection; 5. Model building: survival and clinical data combined with radiomic features.

**Table 1 cancers-13-02522-t001:** Range of the first, second, and higher order textural features.

Order	Features	Examples	Description	Comments
First	Pixel/Voxel Intensity Histogram	Kurtosis, skewness, first-order entropy, mean of all pixels, mean of positive pixels, standard deviation	Gray-level histogram, in which x-axis represents gray level of pixel/voxel and y-axis the frequency of occurrence	Assessment of pixel/voxel intensity without consideration to the relationship with other pixels/voxels
Second	Run-length matrix	Gray-level nonuniformity, run-lenght nonuniformity, long-run emphasis, short-run emphasis	Consecutive pixels/voxels with the same gray level and with a fixed direction	Consider each pixel/voxel intensity and spatial relationships with those adjacent
Grey-level co-occurence matrix	Second-order entropy, sum of entropy, sum of variance, sum of averages	How often occur in an image pairs of pixels with a specified spatial range and specific value	
Higher	Advanced metrics	Geometry parameters, neighborhood gray-tone difference matrix, wavelet energy	Relationships and differences of multiple pixels are compared

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
