# Peer review of "Radiomics in Oncology, Part 1: Technical Principles and Gastrointestinal Application in CT and MRI"

_cancers, 2021, doi:10.3390/cancers13112522_

Round 1
Reviewer 1 Report
This review aims to describe the main technical principles of Radiomics and its upcoming role in gastrointestinal oncologic imaging, with a focus on diagnosis, prediction prognosis, assessment of response to systemic or locoregional therapy.
This review is complete and the division into chapters according to the different organs provides a clear overview of the applications of radiomics.
Abstract
- In my opinion, the sentence “Radiomics could provide quantitative data reflecting microenvironmental tumour heterogeneity and aggressiveness” is not clear. Is heterogeneity a definition correlated with imaging evaluation?
- Please provide an example of limitations of traditional tumour biopsy (sample selection bias, complications, etc).
- authors have used alternatively in the abstract and following sections “diagnosis, prediction prognosis, and assessment of response to therapy in GI cancers” and “cancer detection, prediction prognosis and therapy response assessment”. Indeed diagnosis is very different from tumour detection. Please clarify
Keywords, good.
IntroductionThe sentence “…the biopsy, that often can result too reductive or not diagnostic” needs clarification and stronger references, as it is considered a cornerstone in oncological clinical decision.
Technical PrinciplesThis section is comprehensive, pleasant to read and clear for a non-specialist audience.
I suggest eliminating (or deeply clarify) the mention to the Artificial Intelligence, that is not developed in the paper (line 61) and discarding the Figure 1 that does not provide scientific contribution to the paper.
Acquisition protocols bias is one of the main limitations of clinical use of radiomics. I suggest giving a concise definition of global scaling, z-standardization, and histogram-matching. (line 74).
Authors have stated: “Several options have been recently proposed to overcome the bias of acquisition protocols”. Please expand different advantages and drawbacks according to different clinical scenarios
Feature extraction is the core of radiomics analysis, I suggest improving this section with a figure and/or table aiming at describing detailed information of first/second/high orders features.
I certainly agree that Radiomics features variability is related to segmentation process. Is there a standardized definition of high variability? It is mandatory to know more as high variability features should be discarded.
Esophageal Cancer and Gastric Cancer
Sections are clear and informative.
Liver and Biliary tract
In my opinion, since liver is the only parenchymal organ in this section (among pancreas), it is important to detail the relationship between tumour and surrounding non-tumour parenchyma, and the role radiomics can play in this assessment. In particular, Microvascular Invasion (MVI) is a crucial prognostic factor of hepatocellular carcinoma after surgical resection, and assessment of MVI with classical qualitative imaging is not very effective (Min JH, et al. Interobserver Variability and Diagnostic Performance of Gadoxetic Acid-enhanced MRI for Predicting Microvascular Invasion in Hepatocellular Carcinoma. Radiology. 2020). Two studies are pivotal for the evaluation of peritumoral region in the assessment of MVI at CT and MRI (Xu X, et al. Radiomic analysis of contrast-enhanced CT predicts microvascular invasion and outcome in hepatocellular carcinoma. J Hepatol. 2019; Feng ST, et al. Preoperative prediction of microvascular invasion in hepatocellular cancer: a radiomics model using Gd-EOBDTPA-enhanced MRI. Eur Radiol. 2019).
Pancreas
Reference 45 is not well explained and shouldn’t be in the sentence concerning differential diagnosis between PDAC and pancreatitis (line 278).
Small Bowel, Neuroendocrine Tumors and Colorectal Cancer
Sections are clear and informative.
Figures
Figure 1: see comment above
Figure 2 and 3 are disordered and overlapping, I suggest to remove one figure and to simplify the other, following the path explained in the section on technical principles.
Abbreviations
Please check abbreviation list: for example, absence of MVI and MSI definition.
Author Response
Dear Editor, Dear Reviewers,
We would like to sincerely thank you for the detailed review of our manuscript, your valuable suggestions, and your precise comments.
Please find responses to your comments and consecutive changes based on your recommendations below.
Thank you again.
Reviewer’s comments are in bold. Changes in the manuscript are in italics.
REVIEWER 1
This review aims to describe the main technical principles of Radiomics and its upcoming role in gastrointestinal oncologic imaging, with a focus on diagnosis, prediction prognosis, assessment of response to systemic or locoregional therapy.
This review is complete and the division into chapters according to the different organs provides a clear overview of the applications of radiomics.
Reply: Sincere thanks for your overall positive evaluation of the paper.
Abstract
- In my opinion, the sentence “Radiomics could provide quantitative data reflecting microenvironmental tumour heterogeneity and aggressiveness” is not clear. Is heterogeneity a definition correlated with imaging evaluation?
Reply: We want to thank you for your observation, the sentence has been modified following your suggestion. We hope that the changes performed will make the sentence clear enough.
Please provide an example of limitations of traditional tumour biopsy (sample selection bias, complications, etc).
Reply: Thank you for the precise and valuable comment, we have modified the Abstract by describing the main limitations of tumor biopsy.
- authors have used alternatively in the abstract and following sections “diagnosis, prediction prognosis, and assessment of response to therapy in GI cancers” and “cancer detection, prediction prognosis and therapy response assessment”. Indeed diagnosis is very different from tumour detection. Please clarify
Reply: We apologize for the inappropriateness. We have corrected the Abstract and the following sections accordingly.
Keywords, good.
Reply: Thank you for the positive evaluation.
Introduction
The sentence “…the biopsy, that often can result too reductive or not diagnostic” needs clarification and stronger references, as it is considered a cornerstone in oncological clinical decision.
Reply: We extend to you our most sincere thanks for the comment. We have clarified the sentence and have added some references supporting this key concept.
Technical Principles
This section is comprehensive, pleasant to read and clear for a non-specialist audience.
I suggest eliminating (or deeply clarify) the mention to the Artificial Intelligence, that is not developed in the paper (line 61) and discarding the Figure 1 that does not provide scientific contribution to the paper.
Reply: Thank you for your valuable and precise suggestions, we have proceeded to properly modify and clarify the mention of Artificial Intelligence also by adding a reference (PMID 32742958). In that context, we hope that Figure 1 could provide a graphic insight of Artificial Intelligence approach from a multitude of disorganized data to organized and precise information or bright data. We hope you will agree with us to leave it as it is.
Acquisition protocols bias is one of the main limitations of clinical use of radiomics. I suggest giving a concise definition of global scaling, z-standardization, and histogram-matching. (line 74).
Reply: Thanks for your comment. We have modified the manuscript by adding concise definitions of global scaling, z-standardization, and histogram-matching according to your valuable and helpful comment. We hope that the changes are suitable.
Authors have stated: “Several options have been recently proposed to overcome the bias of acquisition protocols”. Please expand different advantages and drawbacks according to different clinical scenarios
Reply: Text has been improved by describing the different options proposed to overcome the bias of acquisition protocols and two references has been added to support these options (PMID 28112418; 29688159). Thank you for your input.
Feature extraction is the core of radiomics analysis, I suggest improving this section with a figure and/or table aiming at describing detailed information of first/second/high orders features.
Reply: Thank you for your useful comment, we have added a figure (Figure 2) representing a graphical flow-chart of radiomic feature extraction with a specific focus on pre-processing and first/second/high order features. In addition, we have added a table representing features of first, second or higher orders (Table 1).
I certainly agree that Radiomics features variability is related to segmentation process. Is there a standardized definition of high variability? It is mandatory to know more as high variability features should be discarded.
Reply: We apologize for the incomplete definition of features high variability; the text has been improved according to your punctual and valuable suggestion, also by adding references PMID 32728098 and 30838121.
Esophageal Cancer and Gastric Cancer
Sections are clear and informative.
Reply: We extend to you our most sincere thanks for the evaluation.
Liver and Biliary tract
In my opinion, since liver is the only parenchymal organ in this section (among pancreas), it is important to detail the relationship between tumour and surrounding non-tumour parenchyma, and the role radiomics can play in this assessment. In particular, Microvascular Invasion (MVI) is a crucial prognostic factor of hepatocellular carcinoma after surgical resection, and assessment of MVI with classical qualitative imaging is not very effective (Min JH, et al. Interobserver Variability and Diagnostic Performance of Gadoxetic Acid-enhanced MRI for Predicting Microvascular Invasion in Hepatocellular Carcinoma. Radiology. 2020). Two studies are pivotal for the evaluation of peritumoral region in the assessment of MVI at CT and MRI (Xu X, et al. Radiomic analysis of contrast-enhanced CT predicts microvascular invasion and outcome in hepatocellular carcinoma. J Hepatol. 2019; Feng ST, et al. Preoperative prediction of microvascular invasion in hepatocellular cancer: a radiomics model using Gd-EOBDTPA-enhanced MRI. Eur Radiol. 2019).
Reply: Thank you for the comment and the precise suggestion. The text has been improved by adding the valuable references suggested, needed to highlight the role of Radiomics at assessing this relevant prognostic factor for HCC patients.
Pancreas
Reference 45 is not well explained and shouldn’t be in the sentence concerning differential diagnosis between PDAC and pancreatitis (line 278).
Reply: We apologize for the confounding reference 45, it has been removed accordingly.
Small Bowel, Neuroendocrine Tumors and Colorectal Cancer
Sections are clear and informative.
Reply: We are pleased to thank for overall evaluation.
Figures
Figure 1: see comment above
Reply: Thank you for your suggestion, we have modified the text according to the comment above.
Figure 2 and 3 are disordered and overlapping, I suggest to remove one figure and to simplify the other, following the path explained in the section on technical principles.
Reply: We want to thank for the accurate suggestion. We have merged the figures 2 and 3 and add a new figure (new Figure 3) that explain the radiomic path described in the “Technical principles” section.
Abbreviations
Please check abbreviation list: for example, absence of MVI and MSI definition.
Reply: Thank you for your precise suggestion, the abbreviation list has been improved.
I wish all the best,
Sincerely
Andrea Laghi and Co-Authors

Reviewer 2 Report
Thanks for the efforts performed in this manuscript. I have many structural and scientific comments and I hope you accept:
Page 1 Ligne 42 « encrypted medical images »please explain why the medical images are encrypted
Page 2 ligne 55 « incidence and poor prognosis » not all the radiomics studies are focalised on poor prognosis
Page 3 « The next step is the model building, in which the best selected features, clinical data,
patient prognosis and histological data are combined » Please explain what do you mean by patient prognosis in model reconstruction ; patient prognosis is often the objective of the studies.
For Esophageal Cancer, liver an d biliary tract, colorectal, etc it is important not only to specify some of the studies but if the results of one study is confirmed by the others studies.
Colorectal metastasis is probably better to be descibed in liver and biliary tract section .Make specific sections for CHC, CRC, neuroendocrine,etc
Specify , radiomics or radiogenomics in your review and make a separate analysis
For a clinical point of view is important to make a separation betwenn pre therapeutic and treatment assesment/evaluation for each organ and for image modality (CT, MRI,etc)
Author Response
Dear Editor, Dear Reviewers,
We would like to sincerely thank you for the detailed review of our manuscript, your valuable suggestions, and your precise comments.
Please find responses to your comments and consecutive changes based on your recommendations below.
Thank you again.
Reviewer’s comments are in bold. Changes in the manuscript are in italics.
REVIEWER 2
Thanks for the efforts performed in this manuscript. I have many structural and scientific comments and I hope you accept:
Page 1 Ligne 42 « encrypted medical images »please explain why the medical images are encrypted
Reply: Thank you for your precious comment, the text has been modified accordingly.
Page 2 ligne 55 « incidence and poor prognosis » not all the radiomics studies are focalised on poor prognosis
Reply: We apologize for the confounding expression in the text, this has been modified by specifying that the review focused on the main cancer with epidemiological significance.
Page 3 « The next step is the model building, in which the best selected features, clinical data,
patient prognosis and histological data are combined » Please explain what do you mean by patient prognosis in model reconstruction ; patient prognosis is often the objective of the studies.
Reply: Thank you for your valuable suggestion, as you correctly stated, most of the studies have the patient prognosis as endpoint. We are sorry again and we have rewritten the sentence accordingly.
For Esophageal Cancer, liver an d biliary tract, colorectal, etc it is important not only to specify some of the studies but if the results of one study is confirmed by the others studies.
Reply: We want to thank for the accurate suggestion, each section has been modified accordingly.
Colorectal metastasis is probably better to be descibed in liver and biliary tract section .Make specific sections for CHC, CRC, neuroendocrine,etc
Reply: The investigation of liver metastases in CRC has been moved to liver and biliary tract section as you have accurately suggested. However, we have considered to maintain the paper analyzing liver metastases (Martini I. et al.) in neuroendocrine section due to the aim of the study, which investigated textural analysis in differentiating PNET from NPNET. We hope you will accept to leave it as it is, otherwise we will be happy to move this paper from neuroendocrine to liver and biliary tract section.
Specify , radiomics or radiogenomics in your review and make a separate analysis.
Reply: We apologize for the confounding analysis between radiomics and radiogenomics, we have clarified in different sections.
For a clinical point of view is important to make a separation betwenn pre therapeutic and treatment assesment/evaluation for each organ and for image modality (CT, MRI,etc)
Reply: We extend to you our most sincere thanks for the comment, the manuscript has been modified making a separation between pre therapeutic and treatment assessment in each section with the hope to facilitate the reader.
I wish all the best,
Sincerely
Andrea Laghi and Co-Authors

Reviewer 3 Report
In my opinion , all comments have been properly addressed.
Author Response
We extend to you our most sincere thanks for the evaluation.